# A Novel No-Reference Quality Assessment Metric for Stereoscopic Images with Consideration of Comprehensive 3D Quality Information

**DOI:** 10.3390/s23136230

**Published:** 2023-07-07

**Authors:** Liquan Shen, Yang Yao, Xianqiu Geng, Ruigang Fang, Dapeng Wu

**Affiliations:** 1Shanghai Institute for Advanced Communication and Data Science, Shanghai University, Shanghai 200444, China; 2Department of Electrical and Computer Engineering, University of Florida, Gainesville, FL 32603, USA

**Keywords:** no reference, stereoscopic image quality assessment, spatial domain, transform domain, stereo visual information, natural scene statistics, machine learning

## Abstract

Recently, stereoscopic image quality assessment has attracted a lot attention. However, compared with 2D image quality assessment, it is much more difficult to assess the quality of stereoscopic images due to the lack of understanding of 3D visual perception. This paper proposes a novel no-reference quality assessment metric for stereoscopic images using natural scene statistics with consideration of both the quality of the cyclopean image and 3D visual perceptual information (binocular fusion and binocular rivalry). In the proposed method, not only is the quality of the cyclopean image considered, but binocular rivalry and other 3D visual intrinsic properties are also exploited. Specifically, in order to improve the objective quality of the cyclopean image, features of the cyclopean images in both the spatial domain and transformed domain are extracted based on the natural scene statistics (NSS) model. Furthermore, to better comprehend intrinsic properties of the stereoscopic image, in our method, the binocular rivalry effect and other 3D visual properties are also considered in the process of feature extraction. Following adaptive feature pruning using principle component analysis, improved metric accuracy can be found in our proposed method. The experimental results show that the proposed metric can achieve a good and consistent alignment with subjective assessment of stereoscopic images in comparison with existing methods, with the highest SROCC (0.952) and PLCC (0.962) scores being acquired on the LIVE 3D database Phase I.

## 1. Introduction

With the development of entertainment, military and industry, stereoscopic image/video technology has been widely used [1,2,3,4]. Meanwhile, the demand for high-definition and high-quality stereoscopic images/videos has become urgent, which leads to perceptual stereoscopic image quality assessment (SIQA) being an important issue for evaluating the performance of 3D technologies [5]. Stereoscopic images consist of two images, the left image and the right image, which are captured by two cameras [6,7,8]. Two channels of information need to be dealt with simultaneously to assess the quality of stereoscopic images. SIQA is widely used to assess the performance of stereoscopic image processing algorithms, such as stereoscopic image compression and denoising. Furthermore, a good SIQA algorithm can also be utilized to guide stereoscopic image processing, as well as the implementations and optimization proceduresof the image [9].

Intrinsically, stereoscopic image quality is determined not only by the spatial domain factors, but also by the transformed domain factors and stereo perceptual factors, such as binocular disparity/depth perception and binocular rivalry/combination [4,10,11,12,13]. Such dual-image and perceptual factors should be simultaneously considered in the process of feature description in SIQA cases, in addition to cyclopean features. However, almost all the existing feature extraction-based SIQA algorithms have not taken such features into consideration, with only the cyclopean features being exploited. Many assessment methods utilize cyclopean images in a different way. In Chen et al. [14], the natural scene statistics (NSS) model is used to extract spatial features from a cyclopean image, followed by support vector regression (SVR) score fitting. Recently, also based on the generated cyclopean image, Su et al. [15] utilized the spatial domain univariate NSS features, the wavelet domain univariate NSS features, and the bivariate density and correlation NSS features to create another assessment model. In addition, Zhou et al. [16] proposed an assessment model which utilizes the complementary local patterns of binocular energy response (BER) and binocular rivalry response (BRR), both extracted from the cyclopean image, to simulate the binocular visual perception.

In this paper, a perceptual quality assessment metric for stereoscopic images is proposed. The important effort of this work is to take comprehensive dual-image/3D relevant information into consideration of the stereoscopic quality to assess its image quality. Specifically, the traditional goal of an SIQA algorithm is to predict the quality of the cyclopean image formed within an observer’s mind when left and right images are presented stereoscopically. Therefore, like many assessment methods, in our method, features of the cyclopean images in both the spatial domain and transformed domain are extracted based on NSS. Furthermore, to better comprehend the intrinsic properties of the stereoscopic image, in our method, the binocular rivalry effect and other 3D visual properties are also considered in the process of feature extraction. The main contributions of this work are as follows: (1) new 3D perceptual features including binocular rivalry-based NSS features, binocular disparity matching error-based NSS features and binocular disparity consistency-based NSS features, are extracted for binocular perception representation; (2) neighbor difference and neighbor product-based features are optimized to reflect the high correlation between surrounding pixels in the spatial domain to assess the cyclopean image’s quality; (3) gradient magnitude, phase congruency and Log-Gabor response features are designed in the transformed domain to assess the cyclopean image’s quality; and finally, (4) 2D features from the spatial domain and the transformed domain and 3D visual perceptual features are jointly utilized to assess 3D image quality, and principal component analysis (PCA) [17] is utilized to reduce the dimensions of the feature vectors and enhance the performance of the quality prediction.

The proposed paper has several differences from our previously published work, called “No-Reference Stereoscopic Image Quality Assessment Based on Image Distortion and Stereo Perceptual Information” [18]. The differences can be listed as follows: (1) this paper focuses on both 2D and 3D statistics about the given stereoscopic image pairs, such as coherent and rivaling 3D perceptions of the human visual system, while the work in [18] only emphasizes the intrinsic image distortions of 2D image pairs, such as blurriness, noisiness and blockiness; (2) in this work, to exploit more fundamental 3D distortion characteristics, before the machine learning process, a principle component analysis (PCA) is also applied to adaptively recognize the notable features, giving useful attentions for the following machine learning.

The remainder of this paper is organized as follows. Section 2 reviews previous work on SIQAs. We present our model, describing in detail the generation of cyclopean images, the NSS-based feature extraction and quality prediction mechanisms in Section 3. Experimental results are analyzed in Section 4, and finally, conclusions are drawn in Section 5.

## 2. Previous Work on SIQA

Typically, there are two types of stereoscopic image quality assessment (SIQA) methods, which are the subjective SIQA and the objective SIQA. For the subjective one, such assessment is conducted by human observers [19], thus being unstable and time-consuming, hindering its industrial application. Contrarily, the objective SIQA method is conducted by computer algorithms, making it an appropriate solution for industries. Basically, such methods are divided into three types: the full reference (FR) [20,21,22,23,24], the reduced reference (RR) [25,26] and the no reference (NR) [27,28,29,30]. In FR-SIQA methods, evaluations are carried out with both the distorted and the original images provided. In RR-SIQA ones, only partial information from the original image, along with the distorted image, is provided, while in NR-SIQA methods, only the distorted image is given.

For FR-SIQA metrics, four categories altogether can be classified as follows. (1) First is directly applying the traditional image quality assessment algorithms on both the left and right stereo images, and then fusing these scores into the final stereoscopic score [22,23]. Such traditional methods include the peak signal-to-noise ratio (PSNR) [31], the structural similarity index (SSIM) [32] and the MS-SSIM [33]; (2) Second is calculating the tradition IQA scores on both the left and right images and simultaneously taking the stereo perceptual information, such as stereo disparity/depth [34], into consideration. For example, You et al. [35] applied different 2D quality metrics on stereo image pairs as well as the generated disparity map, finding a better combination for good performance. Meanwhile, Benoit et al. [36] computed quality scores of both stereo images and their disparity map, combining them to produce a final score. (3) Thirdly, in addition to evaluating stereo pairs, biological models can be used to simulate binocular combination behaviors [37]. Such metrics usually generate cyclopean images based on the binocular combination model and then evaluate the quality of cyclopean images using 2D-IQA metrics, such as the stereoscopic image quality. For example, Chen et al. [38] formulated binocular rivalry problems by modeling binocular combination behaviors based on a cyclopean map and the estimated disparity map. (4) Fourth is simulating human cognition by building human perception models through the mechanism of the human vision system (HVS) or the binocular visual characteristics, such as binocular fusion, binocular rivalry and binocular suppression [39]. However, due to the original non-distorted reference being missing in most cases, FR-SIQA methods are rarely used in real-world scenarios.

Recently, only a few works have focused on the development of RR and NR-SIQA models. Hewage et al. [40] proposed an RR-SIQA algorithm, in which edges are first computed from depth maps, and then the PSNR between the reference and edge maps is used to analyze the 3D image quality. Akhter et al. [41] proposed an NR-SIQA algorithm, which extracts segmented local features of artifacts from stereo-pairs and the estimated disparity map. In ref. [42], Ryu et al. computed the perceptual blurriness and blockiness scores of left and right images. The two scores are then combined into an overall quality index using the binocular quality perception model. Shao’s methods [27,43] conduct NR tasks without learning based on human opinion scores. Shao’s method [44] specifies the task of SIQA as monocular feature encoding and binocular feature combination based on sparse coding theory. Wang’s scheme [45] extracts features from binocular energy response, and machine learning is used to learn a visual quality prediction model; however, it does not consider any 3D visual characteristics. Fang’s method [46] combines blurriness- and noisiness-based features and information weight features to assess the quality of stereo-pairs. In ref. [47], NSS-based features are extracted from two synthesized images from the cyclopean image and fed into a stacked neural network model for quality prediction.

Recently, with the growth of the deep learning technology, several convolution neural network (CNN)-based methods have been proposed in pursuit of higher performance gains. For example, Du’s method [48] adopts an unsupervised deep network to map images to hierarchical representations. In ref. [49], a general active learning framework is proposed by combining the representativeness and informativeness of samples for active sampling. In ref. [50], an iterative heterogeneous transfer learning framework is proposed that learns a shared feature space for the source and target images and utilizes a novel iterative reweighting strategy to fuse the source samples. Recently, with the advancement of CNNs, several methods also try to implicitly model the feature relations with improved CNN structures. For example, in Zhou et al. [51], a deformable feature extraction block is explored to simulate the function of the primary visual cortex cells, and in ref. [52], a new network block, called the fusion weights assignment module (FWAM), is proposed to adaptively weigh features to guide the fusion of the stereo image features. The solution in [53] analyzes the features using CNN in the wavelet domain. The method in [54] models the relationship between the extracted features and subjective scores using the general regression neural network (GRNN).

Table 1 depicts the considered binocular characteristics and some implementation details of the state-of-the-art SIQA metrics according to the characteristics of HVS. The FR metrics are in italics. “CYC”, “BD”, “BF”, “BR”, “G/P”, “RM”, “SD” and “TD” denote “Cyclopean”, “Binocular Disparity”, “Binocular Fusion”, “Binocular Rivalry”, “Global/Patch”, “Regression Method”, “Spatial Domain” and “Transformed Domain”, respectively. “Global/ Patch” means that the metric is conducted globally across whole images or in patches. “O” and “X” represent “including” and “not including” in the corresponding scheme. The FR metrics are in italics. “NN” denotes a neural network fitting method. It can be seen that the proposed method takes all the listed binocular properties into consideration, while the other metrics lack some of them.

## 3. The Proposed Algorithm

The flowchart of the proposed NR-SIQA method is shown in Figure 1. Binocular disparity is firstly estimated according to an improved-SSIM-based stereo-matching algorithm. Then, a cyclopean image will be generated using stereoscopic images based on a binocular combination model. Next, both 2D NSS features based on neighbor difference and neighbor product in the spatial domain and gradient magnitude, phase congruency and log-Gabor filter responses in the transformed domain are extracted from the cyclopean image. Meanwhile, 3D NSS features are extracted by considering the binocular rivalry, binocular disparity matching error and binocular consistency. Finally, PCA and machine learning are applied for image score computation through training and testing procedures on these extracted feature vectors.

### 3.1. Disparity Estimation

Binocular disparity is an important factor in stereo visual perception and is utilized to reflect the consistency of left and right images. In the past few years, many disparity prediction methods have been proposed, based on the Sum of Absolute Difference (SAD) [38] or SSIM [32]. For SAD-based method, the disparity is predicted by minimizing the SAD between current pixel and its shifted correspondence in the other view. For SSIM-based methods, the disparity is estimated by maximizing the SSIM scores between two such pixels. However, all these methods only achieve satisfactory results when the input quality is high. In this paper, the improved Gaussian average scheme is added to achieve a better SSIM-based disparity estimation performance in low-quality scenarios.

Specifically, both the pixel-matching condition and that for its neighbors are considered when we search one point in other image. For any point in the left image $\left[x_{l},y_{l}\right]$, its corresponding point in the right image is searched within $\left[x_{l}-range,y_{l}\right]$ to $\left[x_{l}+range,y_{l}\right]$, and the range could be changed with specific settings. The searched-for point having the largest calculated similarity with current point in the left image is chosen to be the matched point. Specifically, the SSIM values between those two points are calculated, and all the SSIM values are merged together with a Gaussian weighted sum. The Gaussian weighted sum of SSIM values, considered as the similarity, is used to search for the best matched point. Figure 2 shows the disparity maps of reference stereo-pair ‘im20_l.bmp’ and ‘im20_r.bmp’ from the LIVE 3D IQA database Phase I [55], estimated using the three above-mentioned methods. It can be seen from Figure 2 that the disparity map estimated by the proposed method has less no-texture and disclosure areas, and the rate of error matching of discontinuous areas is also low.

### 3.2. Cyclopean Image

When a human processes the visual stereoscopic images, internally, these signals will be composited into a fusion image, i.e., the cyclopean image, as is analyzed by the binocular vision combination model. To analytically interpret such a process, the Gain-Control Theory model [56] is utilized to conduct binocular fusion and cyclopean perception. Meanwhile, as is suggested in [57], Gabor filter can model the visual processing mechanism of the cells in the primary visual cortex of the human eye in high performance, the normalized Gabor filters are used to weight the paired stereoscopic images. A complex 2D Gabor filter is defined as
(1)$$\begin{array}{cc} \; & GE(x,y,\sigma_{x},\sigma_{y},\xi_{x},\xi_{y},\theta )\\ \; & =\frac{1}{2\pi \sigma_{x}\sigma_{y}}exp\left(-\frac{1}{2}\left[{\left(\frac{R_{1}}{\sigma_{x}}\right)}^{2}+{\left(\frac{R_{2}}{\sigma_{y}}\right)}^{2}\right]\right)exp\left(i\left(x\xi_{x}+y\xi_{y}\right)\right)\\ \; & R_{1}=xcos\theta +ysin\theta \\ \; & R_{2}=-xsin\theta +ycos\theta \end{array}$$
where $\sigma_{x}$ and $\sigma_{y}$ are the standard deviations of an elliptical Gaussian envelope along *x* and *y* axes, $\xi_{x}$ and $\xi_{y}$ are the spatial frequencies of filter and θ denotes the orientation of the filter.

In this paper, the first step of cyclopean image generation is to decide the base image between the left image and the right image. In our implementation, the competitive 2D NR-IQA metric, BRISQUE [58] is utilized to evaluate the quality of the left and right images. The image with better quality is selected as the base image, while the other one is chosen as the aid image.

***Case 1:*** The left image is selected as the base image, and the synthesized cyclopean image I(x,y) is calculated as
(2)$$\begin{array}{cc} \; & I\left(x,y\right)=\omega_{L}\left(x,y\right)\cdot I_{L}\left(x,y\right)\\ \; & \;\;+\omega_{R}\left(x-D_{L}\left(x,y\right),y\right)\cdot I_{R}\left(x-D_{L}\left(x,y\right),y\right) \end{array}$$
where
(3)$$\omega_{L}=\frac{GE_{L}\left(x,y\right)}{GE_{L}\left(x,y\right)+GE_{R}\left(x-D_{L}\left(x,y\right),y\right)}$$
(4)$$\omega_{R}=\frac{GE_{R}\left(x-D_{L}\left(x,y\right),y\right)}{GE_{L}\left(x,y\right)+GE_{R}\left(x-D_{L}\left(x,y\right),y\right)}$$

***Case 2:*** The right image is selected as base image, and the synthesized cyclopean image I(x,y) can be calculated as
(5)$$\begin{array}{cc} \; & I\left(x,y\right)=\omega_{L}\left(x+D_{R}\left(x,y\right),y\right)\cdot I_{L}\left(x+D_{R}\left(x,y\right),y\right)\\ \; & \;\;+\omega_{R}\left(x,y\right)\cdot I_{R}\left(x,y\right) \end{array}$$
where
(6)$$\omega_{L}=\frac{GE_{L}\left(x+D_{R}\left(x,y\right),y\right)}{GE_{L}\left(x+D_{R}\left(x,y\right),y\right)+GE_{R}\left(x,y\right)}$$
(7)$$\omega_{R}=\frac{GE_{R}\left(x,y\right)}{GE_{L}\left(x+D_{R}\left(x,y\right),y\right)+GE_{R}\left(x,y\right)}$$

From Equations (2)–(7), $D_{L}$ or $D_{R}$ is the disparity map estimated by fixing the left or right image and shifting right or left image, respectively. $\omega_{L}$ and $\omega_{R}$ are weighting maps computed using the overall response magnitudes $GE_{L}$ of left image and $GE_{R}$ of right image on all scales and all orientations, respectively.

Figure 3 shows examples of synthesized cyclopean images. The distorted stereoscopic image is asymmetrically distorted by white noise, as shown in Figure 3a,b. Since white noise distortion can increase the right stimulus strength, the cyclopean image in Figure 3c is dominated by the right view with white noise. In the other case, when the stereoscopic image is asymmetrically distorted by Gaussian blur, as shown in Figure 3d,e, the simulated cyclopean image in Figure 3f is determined by the good image, i.e., the left image. This is because blur can decrease the Gabor filter response to a great extent, which causes the weight of the right image to be much smaller than that of the left image. This phenomenon is in accord with the subjective/human perception when stereoscopic images are asymmetrically distorted by different types of distortions.

### 3.3. Feature Extraction

Research on NSS has demonstrated that natural images have certain statistical properties [59], i.e., the probability density distribution of luminance in an image exhibits a Gaussian-like appearance after a local non-linear pre-processing operation called mean subtracted contrast normalization (MSCN) is performed. Applying the local non-linear operation to log-contrast luminances can remove local mean displacements from zero log-contrast and normalize the local variance of the log contrast, thus producing a decorrelating effect [59]. Using these kind of NSS features, several highly competitive 2D NR-IQA models, such as BRISQUE [58], DIIVINE [60] and BLIINDS-II [61], have been developed.

Given a pixel $I_{CC}\left(x,y\right)$ of a cyclopean image, the normalized coefficient $\hat{I_{CC}}\left(x,y\right)$ can be calculated through such an MSCN operation:(8)$$\hat{I_{CC}}\left(x,y\right)=\frac{I_{CC}\left(x,y\right)-\mu \left(x,y\right)}{\sigma (x,y)+C}$$
where *C* is a small constant to increase the stability of $\hat{I_{CC}}\left(x,y\right)$ when the denominator tends to zero. In our implementation, a 7×7 Gaussian weighting matrix with a sigma of 1.17 pixels is utilized to compute μ and σ, and *C* is fixed to 1.

A zero-mean generalized Gaussian distribution (GGD) model is used to fit the probability density distributions of $\hat{I_{CC}}\left(x,y\right)$ as
(9)$$f\left(x;\alpha ,\sigma^{2}\right)=\frac{\alpha }{2\beta \Gamma (1/\alpha )}exp\left(-{\left(\frac{|x|}{\beta }\right)}^{\alpha }\right)$$
where $\beta =\sigma \sqrt{\frac{\Gamma \left(1/\alpha \right)}{\Gamma \left(3/\alpha \right)}}$ and Γ(·) is the gamma function.

The shape parameter α controls the real ‘shape’ of the distribution, while $\sigma^{2}$ controls the variance. The MSCN operation can show the statistics properties of images with different shape, kurtosis and spread in the presence of distortion. This means that it is possible to predict the type of distortion as well as its perceptual quality through quantifying these changes. The parameters α and $\sigma^{2}$ are effective features, which can be reliably estimated using the moment-matching-based approach in [62].

It also can be found that statistical relationships exist among neighboring pixels along different orientations [58]. The empirical distribution of the statistical relationships between pixels along different orientations is not logically symmetric about zero. To better fit the empirical histograms of the relationships, the very general asymmetric generalized Gaussian distribution (AGGD) model [63] is adopted. The AGGD with zero mode is given by
(10)$$\begin{array}{cc} \; & f\left(x;v,\sigma_{l}^{2},\sigma_{r}^{2}\right)=\left\{\begin{array}{cc} \; & \frac{v}{(\beta_{l}+\beta_{r})\Gamma \left(1/v\right)}exp\left(-{\left(\frac{-x}{\beta_{l}}\right)}^{v}\right),x<0\\ \; & \frac{v}{(\beta_{l}+\beta_{r})\Gamma \left(1/v\right)}exp\left(-{\left(\frac{-x}{\beta_{r}}\right)}^{v}\right),x\geq 0 \end{array}\right. \end{array}$$
where $\beta_{l}=\sigma_{l}\sqrt{\frac{\Gamma \left(1/v\right)}{\Gamma \left(3/v\right)}}$ and $\beta_{r}=\sigma_{r}\sqrt{\frac{\Gamma \left(1/v\right)}{\Gamma \left(3/v\right)}}$

The shape parameter *v* controls the real ‘shape’ of the distribution, while $\sigma_{l}^{2}$ and $\sigma_{r}^{2}$ are scale parameters that control the variance on each side of the mode. The parameters of the AGGD distribution are also estimated using the moment-matching-based approach in [62]. All four parameters $\left(\eta ,v,\sigma_{l}^{2},\sigma_{r}^{2}\right)$ of AGGD are extracted as features where η is computed by
(11)$$\eta =\left(\beta_{r}-\beta_{l}\right)\frac{\Gamma \left(2/v\right)}{\Gamma \left(1/v\right)}$$

Inspired by the success of NSS in image quality assessment, a large amount of images with different contents and different distortions are selected from the LIVE database [64], and a series of experiments are conducted based on these images to prove that there is identical statistical regularity between different contents of images and that the distortions will remarkably disturb the statistical distributions of the images. The reference (REF) image “*building2.bmp*” and its five distorted versions, including such distortions as jpeg2000 (JP2K), JPEG, white noise (WN), Gaussian blur (BLUR) and fast fading (FF), are chosen as examples in Figure 4 and Figure 5a–c to show the experimental results. The NSS feature extraction and explanation of Figure 4 and Figure 5a–c are specified as follows.

#### 3.3.1. Neighbor Difference-Based 2D Features

There is high correlation between surrounding pixels of an image in the spatial domain [58]. The difference between neighboring pixels and the product of neighboring pixels can effectively capture information on the image quality.

The difference between adjacent MSCN coefficients $\hat{I_{CC}}\left(x,y\right)$ of a cyclopean image are computed at a distance of one pixel along four orientations: horizontal (H), vertical (V), main-diagonal (D1) and secondary-diagonal (D2), which is illustrated in Figure 6. Specifically,
(12)$$\begin{array}{cc} \; & dif_{H}\left(i,j\right)=\hat{I_{CC}}\left(i,j\right)-\hat{I_{CC}}\left(i,j+1\right)\\ \; & dif_{V}\left(i,j\right)=\hat{I_{CC}}\left(i,j\right)-\hat{I_{CC}}\left(i+1,j\right)\\ \; & dif_{D1}\left(i,j\right)=\hat{I_{CC}}\left(i,j\right)-\hat{I_{CC}}\left(i+1,j+1\right)\\ \; & dif_{D2}\left(i,j\right)=\hat{I_{CC}}\left(i,j\right)-\hat{I_{CC}}\left(i+1,j-1\right) \end{array}$$
where i∈{2,...M−1} and j∈{2,...N−1} and M,N are the image width and height, respectively.

Figure 4a shows the probability density distributions of neighbor differences along the horizontal orientation. Neighbor differences along these four orientations are parameterized using the GGD model to effectively capture the image statistics in the presence of distortion. By estimating two parameters, α and $\sigma^{2}$, in each orientation as the spatial domain features, eight features are obtained overall.

#### 3.3.2. Neighbor Product-Based 2D Features

In Figure 7, we also model the statistical correlation of pairwise products of neighbor pixels at a distance of two pixels along eight orientations, i.e., $0^{\circ }$, $22.5^{\circ }$, $45^{\circ }$, $67.5^{\circ }$, $90^{\circ }$, $112.5^{\circ }$, $135^{\circ }$ and $157.5^{\circ }$. Specifically,
(13)$$\begin{array}{cc} \; & pro_{1}\left(i,j\right)=\hat{I_{CC}}\left(i,j\right)\cdot \hat{I_{CC}}\left(i,j+2\right)\\ \; & pro_{2}\left(i,j\right)=\hat{I_{CC}}\left(i,j\right)\cdot \hat{I_{CC}}\left(i+1,j+2\right)\\ \; & pro_{3}\left(i,j\right)=\hat{I_{CC}}\left(i,j\right)\cdot \hat{I_{CC}}\left(i+2,j+2\right)\\ \; & pro_{4}\left(i,j\right)=\hat{I_{CC}}\left(i,j\right)\cdot \hat{I_{CC}}\left(i+2,j+1\right)\\ \; & pro_{5}\left(i,j\right)=\hat{I_{CC}}\left(i,j\right)\cdot \hat{I_{CC}}\left(i+2,j\right)\\ \; & pro_{6}\left(i,j\right)=\hat{I_{CC}}\left(i,j\right)\cdot \hat{I_{CC}}\left(i+2,j-1\right)\\ \; & pro_{7}\left(i,j\right)=\hat{I_{CC}}\left(i,j\right)\cdot \hat{I_{CC}}\left(i+2,j-2\right)\\ \; & pro_{8}\left(i,j\right)=\hat{I_{CC}}\left(i,j\right)\cdot \hat{I_{CC}}\left(i+1,j-2\right) \end{array}$$
where i∈{3,...M−2} and j∈{3,...N−2} and M,N are the image width and height, respectively.

Figure 4b shows examples of the probability density distributions of neighbor products along the horizontal orientation, i.e., $pro_{1}\left(i,j\right)$. We utilize the AGGD model to fit the empirical histograms of the paired neighbor products, and the parameters $\left(\eta ,v,\sigma_{l}^{2},\sigma_{r}^{2}\right)$ are estimated as the spatial domain features. Finally, 32 features are extracted for these 8 paired products.

#### 3.3.3. Gradient Magnitude-Based 2D Features

As suggested in [65], image gradient magnitudes (GM) can reflect some properties of images. Specifically, GM is computed to be the root mean square of the directional gradients in the horizontal and vertical directions. In this paper, the Scharr operators [66] are applied along the horizontal (*x*) and vertical (*y*) directions, which are defined as
(14)$$h_{x}=\frac{1}{16}\left[\begin{array}{ccc} 3 & 0 & -3\\ 10 & 0 & -10\\ 3 & 0 & -3 \end{array}\right],h_{y}=\frac{1}{16}\left[\begin{array}{ccc} 3 & 10 & 3\\ 0 & 0 & 0\\ -3 & -10 & -3 \end{array}\right]$$

Convolving $h_{x}$ and $h_{y}$ with cyclopean image $I_{CC}$ produces the horizontal gradient $GM_{x}\left(x\right)$ and vertical gradient $GM_{y}\left(y\right)$. The gradient magnitude of the cyclopean image is defined as
(15)$$\begin{array}{cc} \; & GM\left(x\right)=\sqrt{{\left(I_{CC}⊗h_{x}\right)}^{2}+{\left(I_{CC}⊗h_{y}\right)}^{2}}\\ \; & \;\;=\sqrt{GM_{x}^{2}\left(x\right)+GM_{y}^{2}\left(y\right)} \end{array}$$
where the symbol “⊗” denotes the convolution operation.

Figure 4c shows the probability density distributions of $GM_{y}\left(y\right)$ after the MSCN operation. The GGD model is utilized to fit the distributions of GM(x), $GM_{x}\left(x\right)$ and $GM_{y}\left(y\right)$ after MSCN, and the parameters α and $\sigma^{2}$ are estimated as the NSS-based transformed domain features.

#### 3.3.4. Phase Congruency-Based 2D Features

Phase congruency (PC) reflects the image changes in brightness or contrast, which represents the spatial and phase locations where the maximum Fourier response is received [67]. This can be considered as the coherency degree of the image’s local frequencies. In this paper, PC is computed by Kovesi’s salient work [67] on the cyclopean image.

For cyclopean image $I_{CC}$, the multi-direction and multi-scale complex Gabor filters are applied. $M_{n}^{e}$ and $M_{n}^{o}$ denote the even-symmetric (cosine) and odd-symmetric (sine) filters at scale *n*, respectively, forming a quadrature pair. By convolving $I_{CC}$ with these filters, the responses will form a vector at position *x*, on scale *n* and orientation θ:(16)$$\left[e_{n,\theta }\left(x\right),o_{n,\theta }\left(x\right)\right]=\left[I_{CC}⊗M_{n}^{e},I_{CC}⊗M_{n}^{o}\right]$$
where the symbol “⊗” denotes the convolution operation and θ is the orientation angle of the filters. The local amplitude of point *x* on scale *n* and orientation θ is computed by
(17)$$A_{n,\theta }\left(x\right)=\sqrt{e_{n,\theta }^{2}\left(x\right)+o_{n,\theta }^{2}\left(x\right)}.$$
The local energy along orientation θ is computed by
(18)$$E_{\theta }\left(x\right)=\sqrt{F_{\theta }^{2}\left(x\right)+H_{\theta }^{2}\left(x\right)}$$
where $F_{\theta }\left(x\right)=\sum_{n}e_{n,\theta }\left(x\right)$, $H_{\theta }\left(x\right)=\sum_{n}o_{n,\theta }\left(x\right)$. Then, the phase congruency of the cyclopean image at point *x* is defined by
(19)$$PC\left(x\right)=\frac{\sum_{j}E_{\theta }\left(x\right)}{\epsilon +\sum_{n}\sum_{j}A_{n,\theta }\left(x\right)}$$
where ϵ is a small positive constant, *j* is the orientation angle number and *n* is the scale number.

Figure 5a shows the probability density distributions of the phase congruency map after MSCN is performed. To extract features from the phase congruency map, the AGGD model is used to fit the MSCN coefficients of phase congruency and AGGD model parameters $\left(\eta ,v,\sigma_{l}^{2},\sigma_{r}^{2}\right)$ are extracted as the transformed domain features.

#### 3.3.5. Log-Gabor Response-Based 2D Features

Since neurons in the visual cortex selectively respond to stimuli in different orientations and frequencies, filters with multiple scales and multiple orientations are useful to simulate this physiological phenomenon [57]. These filters can closely model frequency–orientation decompositions in the primary visual cortex and can capture energy in a highly localized manner in both space and frequency domains [68]. Such filter responses of an image are also useful for generating quality-aware NR-IQA features. Here, perceptually relevant log-Gabor filters are exploited to accomplish multi-scale and multi-orientation filtering.

A 2D log-Gabor filter with multiple frequencies and multiple orientations in the Fourier domain can be expressed as
(20)$$G_{log}\left(\omega ,\theta \right)=exp\left(-\frac{{\left(log(\omega /\omega_{0})\right)}^{2}}{2\sigma_{r}^{2}}\right)\cdot exp\left(-\frac{{\left(\theta -\theta_{j}\right)}^{2}}{2\sigma_{\theta }^{2}}\right)$$
where $\theta_{j}=j\pi /J,j=\left\{0,1,...J-1\right\}$ is the orientation of the log-Gabor filter, *J* is the overall number of orientations, $\omega_{0}$ is the center frequency of the filter, $\sigma_{r}$ determines the filter’s radial bandwidth and $\sigma_{\theta }$ controls the filter’s angular bandwidth. After convolving with the cyclopean image $I_{CC}$, the local responses of the real part, $e_{n,\theta_{j}}\left(x\right)$, and imaginary part, $o_{n,\theta_{j}}\left(x\right)$, of the log-Gabor filter are obtained at position *x* on scale *n* and orientation $\theta_{j}$.

The log-Gabor filter response of cyclopean image $I_{CC}$ is expressed as
(21)$$G_{log}\left(x\right)=\sum_{n}\sum_{j}\sqrt{e_{n,\theta_{j}}^{2}\left(x\right)+o_{n,\theta_{j}}^{2}\left(x\right)}.$$

The real and imaginary parts of the log-Gabor response of cyclopean image $I_{CC}$ are respectively expressed as follows:(22)$$\begin{array}{cc} \; & Gx_{log}\left(x\right)=\sum_{n}^{}\sum_{j}^{}e_{n,\theta_{j}}^{2}\left(x\right)\\ \; & Gy_{log}\left(x\right)=\sum_{n}^{}\sum_{j}^{}o_{n,\theta_{j}}^{2}\left(x\right) \end{array}$$

The log-Gabor filter phase of cyclopean image $I_{CC}$ is expressed as
(23)$$GP_{log}\left(x\right)=\sum_{n}^{}\sum_{j}^{}arctan\left(e_{n,\theta_{j}}\left(x\right),o_{n,\theta_{j}}\left(x\right)\right)$$

Figure 5b,c show the probability density distributions of $Gx_{log}\left(x\right)$ and $GP_{log}\left(x\right)$ after MSCN operation, respectively. The probability density distributions of $G_{log}\left(x\right)$, $Gx_{log}\left(x\right)$, $Gy_{log}\left(x\right)$ and $GP_{log}\left(x\right)$ after MSCN is performed can be fitted using the model of GGD, and the parameters α and $\sigma^{2}$ are included in the feature set.

#### 3.3.6. Binocular Rivalry-Based 3D Features

Binocular disparity or depth is an important factor in stereo visual perception. Binocular disparity is computed by the distance between the corresponding pixels in the left and right images. The binocular disparity can be regarded as the measurement that the human brain senses the depth information with. In general, it can lead to binocular rivalry and binocular suppression. As indicated by [35,38], the disparity or depth information has an impact on stereoscopic image quality. Therefore, the binocular disparity can be used to generate the binocular rivalry properties and enhance the detection accuracy of the saliency map.

According to visual psychophysical research, binocular combination will integrate two retinal points into a single binocular perception when similar contents of the left and right images fall on the corresponding retinal points [69]. Binocular rivalry happens when the content of the left and right images is relatively different. Therefore, the left and right eyes view mismatched images at the same retinal locations. Numerous studies suggest that binocular rivalry is the result of competition between human eyes [70]. This competition is caused by the mismatched content of the left and right images. Binocular suppression is a special case of binocular rivalry. When binocular suppression happens, the entire image from one retina is partly suppressed [1].

We use the binocular disparity estimated in Section 3.1 to describe the binocular rivalry. A variety of stereo-pairs with different distortions and contents are selected from the LIVE 3D database Phase I [55] to validate the effectiveness of the binocular rivalry-based 3D features and the following 3D features. The statistical distributions of the stereo-pair “im17_l.bmp” and “im17_r.bmp” are chosen as an example in Figure 5d–f. Figure 5d shows the probability density distributions of the binocular disparity map after MSCN operation. We use parameters α and $\sigma^{2}$ of the GGD model to represent the statistical features of binocular disparity.

#### 3.3.7. Binocular Disparity Matching Error-Based 3D Features

Except for the estimated binocular disparity, the disparity matching error caused by the improved SSIM-based stereo matching algorithm is also a useful feature related to the stereo image quality. One pixel fails to find the corresponding pixel in the other image in a stereo-pair when large matching error exists. The disparity matching error is defined as follows:(24)$$\begin{array}{cc} \; & D_{L}error\left(x,y\right)=I_{L}\left(x,y\right)-I_{R}\left(x-D_{L}\left(x,y\right),y\right)\\ \; & D_{R}error\left(x,y\right)=I_{L}\left(x+D_{R}\left(x,y\right),y\right)-I_{R}\left(x,y\right) \end{array}$$
where $D_{L}$ and $D_{R}$ are disparity maps estimated using the left and right images as the base image, respectively. $I_{L}\left(x+D_{R}\left(x,y\right),y\right)$ and $I_{R}\left(x-D_{L}\left(x,y\right),y\right)$ are the left and right disparity-compensated images, respectively.

Figure 5e shows the probability density distribution of binocular disparity matching error map after MSCN operation. Then, the GGD model is utilized to fit the probability density distribution, and the parameters $\left(\alpha ,\sigma^{2}\right)$ are estimated as 3D features.

#### 3.3.8. Binocular Disparity Consistency-Based 3D Features

Like normal natural images, there is high correlation between neighboring pixels, except in some edge regions in a disparity map. Thus, one point in a disparity map is usually similar to its surrounding points when the disparity map is computed from an original, undistorted stereoscopic image. We describe the disparity consistency by using a high-pass filter to convolve with the estimated disparity map D(x,y). The disparity consistency is defined as
(25)$$D_{consist}\left(x,y\right)=D\left(x,y\right)⊗\left[\begin{array}{ccc} 0 & \frac{1}{4} & 0\\ \frac{1}{4} & -1 & \frac{1}{4}\\ 0 & \frac{1}{4} & 0 \end{array}\right]$$

The value of $D_{consist}\left(x,y\right)$ would tend to zero if the stereoscopic image has no distortion or little distortion. Figure 5f shows the probability density distributions of disparity consistency maps after MSCN operation. These six corresponding disparity maps are extracted from the stereo-pair im17.bmp and its five distorted versions from the LIVE 3D database Phase I. It can be seen that the coefficient values from a disparity consistency map mostly concentrate to zero. However, when the disparity is obtained from a distorted stereo-pair, the coefficient values of its disparity consistency map are more likely to spread out. In addition, the probability of coefficient values equal to zero is much lower.

The GGD model is used to fit the empirical distribution of $D_{consist}\left(x,y\right)$ after the MSCN operation is performed, and the GGD parameters $\left(\alpha ,\sigma^{2}\right)$ are extracted as features.

### 3.4. Machine Learning

To adaptively choose a salient feature set for better regression model fitting, the principal component analysis (PCA) is applied, and the chosen salient features are sent into the support vector regressor (SVR) for the final fitting of the objective quality score. In the training stage, the generated features and the labeled subjective quality scores are sent to the regression model, and the training epoch is 1000. In the testing stage, the predicted objective scores are obtained from the trained regressor. Our implementation is based on the MATLAB platform, and the SVR is conducted using the LIBSVM package [2]. For the SVR’s settings, the epsilon-SVR with a radial basis function (RBF) kernel is used, and the SVR parameters *C* and γ are set to 512 and 0.015625, respectively, according to extensive experiments. The details about the parameter selection can be found in Section 4.2.

## 4. Experimental Results and Analysis

### 4.1. Databases and Evaluation Criteria

For training and testing datasets, we adopt the LIVE 3D IQA database Phase I [1] and Phase II [14] and the Waterloo IVC (UW/IVC) 3D IQA database [3] for testing the proposed algorithm and other SIQA algorithms. For LIVE 3D Phase I, 20 reference stereoscopic images and 365 distorted stereoscopic pairs are included, each having 5 types of distortions, i.e., JP2K compression distortion, JPEG compression distortion, white noise (WN), fast fading (FF) and blurring (BLUR). All distortions are symmetric in nature. Similarly, for LIVE 3D Phase II, 8 reference stereoscopic images, 120 symmetrically distorted stereoscopic pairs and 240 asymmetrically distorted stereoscopic pairs generated from those 8 references are included. The Waterloo IVC 3D IQA database also contains two data phases, created from 6 and 10 pristine stereo-pairs, respectively. Both of these phases include symmetrical and asymmetrical distortions, including additive white Gaussian noise, Gaussian blur and JPEG compression. Each type of distortion has four levels. Altogether, a total of 78 single-view images and 330 stereoscopic images are contained in Phase I, and a total of 130 single-view images and 460 stereoscopic images are included in Phase II.

To evaluate the overall accuracy of the stereoscopic objective assessment, three criteria are used: the Pearson linear correlation coefficient (PLCC), the Spearman rank order correlation coefficient (SROCC) and the root mean squared error (RMSE). Traditionally, the PLCC and RMSE are used to evaluate the prediction accuracy of SIQA metrics, and the SROCC detects the prediction’s monotonicity. Note that the higher the SIQA’s accuracy, the higher the values PLCC and SROCC will be, while the value of the RMSE should be lower. Before the evaluation, a non-linear regression analysis, suggested by the video quality experts group [4], is utilized to provide mappings between the objective scores and subjective mean opinion scores (MOSs). Specifically, a five-parameter logistic function [10] is adopted as such a non-linear mapping:(26)$$f\left(x\right)=\beta_{1}\cdot \left(\frac{1}{2}-\frac{1}{1+e^{\beta_{2}\left(x-\beta_{3}\right)}}\right)+\beta_{4}\cdot x+\beta_{5}$$
where $\beta_{i}=1,2,...,5$ are parameters determined by the subjective scores and the objective scores.

### 4.2. Implementation Details

The implementation of the whole procedure is conducted on MATLAB, and the support vector regressor is conducted using LibSVM. The PCA is introduced and its effectiveness is validated by various experiments. The following experiments are also conducted to select the SVR parameters (*C*, γ) and prove that almost no over-fitting exists in our method.

#### 4.2.1. PCA for Feature Dimension Reduction

In the proposed algorithm, more than 60 features are extracted from the 2D cue and 3D cue for each image. Inevitably, information redundancy may be contained in these NSS features (i.e., the shape feature). To highlight the main component and reduce the dimensions of the feature vectors, PCA is utilized, resulting in a more efficient quality predictor. Let $X=\left[x_{1},x_{2},...,x_{n}\right]\in {ℜ}^{d\times n}$ denote the feature matrix extracted from *n* images, and *d* is the number of features extracted from each image. By applying PCA to X, a transform matrix, $\Phi \in {ℜ}^{d\times m}$, will be obtained by an eigenvalue decomposition conducted on the covariance matrix of the feature matrix X. The transform matrix is formed by the m(m<d) principle vectors associated with the *m* most significant eigenvalues of the covariance matrix of X. Then, the dimension of each feature vector $x_{i}$ can be reduced by transformation as follows:(27)$$x_{i}^{\prime }=\Phi^{T}x_{i},x_{i}^{\prime }\in {ℜ}^{m\times 1}\;i=1,2,...,n$$

The PLCC with various PCA dimensions in LIVE Phase I and II is shown in Figure 8. As the number of feature dimensions is set to 44, the performance is the best. *m* is set to 44 in our implementation.

#### 4.2.2. SVR Parameters (*C*, γ) Selection

A cross-validation experiment is conducted to choose the value of (*C*,γ). During this experiment, samples from LIVE 3D Phase I and II are randomly partitioned into two sets: the training set and the testing set. For each phase, 80% of the images are for training and the remaining 20% of the images are for testing. The training epoch is 1000, and the median results are reported for performance evaluation. Parameters (*C*,γ) delivering the best median result are chosen as the best. The illustration of this experimental process can be found in Figure 9. The number of the level contour indicates the SROCC values for the cross-validation. Apparently, there exists a circular region surrounding the center, where the proposed metric obtains the best SROCC results on the two databases. This indicates that this learning model is more robust to different databases of samples. According to the results, the optimal parameters (*C*,γ) are set to be 512, 0.015625.

#### 4.2.3. Two-Fold Cross-Validation

Each database is randomly split into two sets: the training set and the testing set, with a 50:50 splitting ratio. Performance evaluations are conducted as follows. First, the optimal regression model is learned from the training set, and then tested on the training set and testing set, respectively. Next, the other regression model is learned from the testing set, and then tested on the training set and testing set, respectively. Performance evaluations on LIVE Phase I and II are listed in Table 2 (“MTrain→Testing set” means that the model is learned from the training set is tested on the testing set, and “MTest→Training set” represents that the model learned from the testing set is tested on the training set). As is seen on the table, regardless of whether the regression model is learned from the training set or the testing set, no obvious difference exists between the evaluated performance, indicating that almost no over-fitting exists in the proposed metric.

#### 4.2.4. Features with Noise-Addition

Here, we test the impact of Gaussian white noise (WN) on the model’s fitting performance. Similarly, LIVE 3D Phase I and II are randomly partitioned into the training set and the testing set with the ratio 80:20. All the features of the samples in the training set are applied with white noise before the experiment. The variance of the white noise is set to 0.01. Performance evaluations are conducted on the training set affected or not affected by white noise, and then tested on the non-affected training set and testing set, respectively. The results are listed in Table 3, where “WN→Training set” means that the model is learned from the WN-affected training set and is tested on the non-affected training set, and “WN/F→Testing set” denotes that the model is learned from the non-affected training set and is tested on the non-affected testing set. As is seen in the table, there is no big difference between the model’s performance when tested on the training set or testing set. Moreover, the results also indicate that white noise has barely any impact on the model’s fitting process. Therefore, we can make a conclusion that almost no over-fitting exists in the proposed metric.

### 4.3. Performance Comparison

In the proposed method, 80% images of each database are chosen randomly as training set and the rest is for testing. The training-testing process is repeated 1000 times, and the average of performance is reported for comparison.

#### 4.3.1. Overall Performance Comparison

The overall performance comparisons of the proposed algorithm and other NR-SIQA methods on both LIVE Phase I and II are shown in Table 4. Shao [27,44] split the LIVE Phase II into symmetrically and asymmetrically distorted dataset, and thus the overall performance on whole LIVE Phase II is unavailable, which represented by “-”. It can be seen that the proposed method achieves much higher performance than other state-of-the-art metrics. Besides, it also can be seen from Table 4 that the proposed metric and other NR-SIQA metrics achieve a better performance on Phase I than those on Phase II. The reason is that Phase I only contains symmetric distortions, while Phase II contains both symmetric and asymmetric distortions. It is much more difficult to evaluate the stereoscopic image quality with asymmetric distortions due to the limited understanding of HVS when human watch asymmetrically distorted stereoscopic images.

Table 5 presents the comparison results between the proposed method and other FR and NR metrics on Waterloo IVC 3D database. In Table 5, the FR methods are marked in italic. Specifically, Mittal et al. [58] is a 2D NR metric; therefore, its performance is evaluated on the both left and right views, resulting two objective scores. The final score for this method is the average of these two scores. It can be seen that the proposed method outperforms the other metrics. The reason that Chen’s [14] method performs worse than Mittal’s [58] may be because its stereo-matching algorithm fails to generate precise disparity.

#### 4.3.2. Performance Comparison of Each Distortion Type

Here, we discuss the performance comparisons over different distortions. Several state-of-the-art FR-SIQA metrics (i.e., Benoit [36], You [35], Gorley [22] and Chen [38]), along with our method, are tested on LIVE Phase I and II, and the results are listed in Table 6. Note that the best scores are marked in bold in that table. According to the table, the proposed algorithm achieves a slightly higher performance than the algorithm in [44], while performing better than other FR-SIQA and NR-SIQA metrics. As FR-SIQA metrics, Gorley et al. [22] predicts the overall quality by averaging the scores of left and right views generated from 2D IQA algorithms. Its performance is much worse than other SIQA methods because it does not take the binocular visual properties into account. In addition to the dual-2D quality scores for both views, You et al. [35] and Benoit et al. [36] simultaneously consider disparity information, and the performances of these two methods are better than Gorley et al., as their SROCC and PLCC scores reach up to 0.90 on LIVE Phase I. Chen et al.’s model [38] applies the SSIM to assess the quality of cyclopean images, making its SROCC and PLCC scores reach up to 0.920 on LIVE Phase I. For NR-SIQA metrics, Shao et al. [27] may be opinion-unaware; however, its performance is inferior to other methods based on prior subjective human opinion scores. Shao et al. [44] achieves better performance than the other methods, except for the proposed one, as it lacks the binocular properties of our method. The performance of method [41] is barely satisfactory on JP2K and JPEG distortion in the LIVE Phase I database. For Chen et al.’s method [14], it extracts NSS-based features of the cyclopean image in the spatial domain, but it lacks considerations of the binocular rivalry effect and other 3D visual perceptions. Its performance is worse than our proposed algorithm, as our method sufficiently extracts 2D and 3D features, and takes 3D visual characteristics into consideration.

### 4.4. Performance Evaluation

#### 4.4.1. Performance Evaluation of Each Feature Type

In our metric, features including neighbor difference (ND)-based NSS features and neighbor product (NP)-based NSS features in the spatial domain and gradient magnitude (GM)-based NSS features, phase congruency (PC)-based NSS features, log-Gabor filter response (GR)-based NSS features in the transformed domain and 3D information (3DI), such as binocular rivalry, binocular disparity matching error and binocular disparity consistency NSS features, are extracted. In order to verify whether some of these features could be ignored, we analyze the gain of each type of feature in Table 7. Note that this experiment is conducted with an incomplete feature set; thus, the PCA process is not used to reduce the feature dimensions. We can see from Table 7 that each type of feature can improve the performance of our model.

#### 4.4.2. Cross-Database Performance Evaluation

The evaluation strategy used in Section 4.3 is inadequate to evaluate the generalization capability of a NR-SIQA model, since the random partition method on a single database leads to the training set and the testing set containing the same types of distortions. Therefore, to demonstrate the efficiency of an IQA model on other possibly unknown distortions, and to verify such methodologies are actually “completely blind”, we test the generalization capability of the proposed method by training the regression model on one database, i.e., LIVE Phase I, and testing the regression model on another database, i.e., LIVE Phase II.

In our implementation, we train the regression model on LIVE Phase I and test it on LIVE Phase II on one hand, and on the other hand, we train the model on LIVE Phase II and test it on LIVE Phase I. The results are reported in Table 8 and Table 9, from which we can draw the conclusion that the regression model that is trained on one database (i.e., LIVE Phase I) is not completely efficient for testing on the other database (i.e., LIVE Phase II), since the contents in the two databases are quite different. Phase II includes both symmetrically and asymmetrically distorted stereo-pairs, while Phase I only contains symmetrically distorted stereo-pairs.

#### 4.4.3. Cross-Distortion Performance Evaluation

Here, we verify the cross-distortion performance comparisons between the proposed method and the other methods. Specifically, each tested method is trained on one type of distortion but tested on another type, with the same training sample sources. The results are displayed in Table 10 and Table 11. According to the row elements of the tables, we can see that the model trained on one type of distortion has some influence on the performance on other types of distortion, but with less influence on the overall performance. This demonstrates that the proposed scheme is a good metric for the perceived quality prediction of the stereoscopic image.

#### 4.4.4. Image Content-Based Performance Evaluation

In this experiment, we discuss the database partitioning strategies, which are split by content or split randomly. During this experiment, we took 80% of the samples for training and the rest for testing. Specifically, for the LIVE 3D database Phase I, 16 reference images and their corresponding distorted images were selected as the training sets, while the others were chosen as the testing sets. For LIVE 3D Phase II, six samples were chosen for training, and the other two were used for testing. The comparison results between the content-based partitioning and random partitioning are listed in Table 12. It can be seen that the performance using image content-based partitioning is slightly worse than that using random partitioning. This is because the regression model learned from the content-based partitioned samples is a “content-aware” model, and thus the performance is a little worse than that of the “content-unaware” model.

### 4.5. Influence of Disparity Estimate Algorithm

In order to verify whether our metric is method-independent on disparity estimation, we compare the performance of the improved SSIM-based disparity search algorithm in our method with those of the SAD-based and SSIM-based disparity estimation algorithms. The performance comparisons are listed in Table 13. It can be seen from Table 13 that different disparity estimation algorithms lead to similar performance, which means our metric is method-independent. It also can be seen that the performance from the improved SSIM-based disparity search algorithm is slightly better.

### 4.6. Complexity Analysis

The time–cost ratios for each step of our algorithm are listed in Table 14. As is described in the table, the biggest share of complexity in our method is from the PCA ans SVM fitting steps, which cost more than half of our prediction time, followed by the SSIM matching step that costs 36% of our prediction time.

## 5. Conclusions

In this paper, we propose a novel natural scene statistics-based, no-reference quality assessment algorithm for stereoscopic images, in which both the quality of the cyclopean image is considered and the binocular rivalry and other 3D visual intrinsic properties are exploited. The proposed algorithm mainly consists of two steps. First, 2D features are extracted from the cyclopean image in the spatial domain and transformed domain, and then 3D quality-relevant perceptual features are extracted from stereo visual perceptual information, including the binocular rivalry, the disparity matching error and the disparity consistency. Next, followed by NSS modeling and adaptive PCA feature pruning, SVR is utilized to fit an optimum regression model. Compared with other FR-SIQA metrics and NR-SIQA metrics on the LIVE Phase I and LIVE Phase II databases, the proposed stereoscopic image quality metric achieves the best performance compared to other state-of-the-art SIQA metrics.

## Figures and Tables

**Figure 1 sensors-23-06230-f001:**
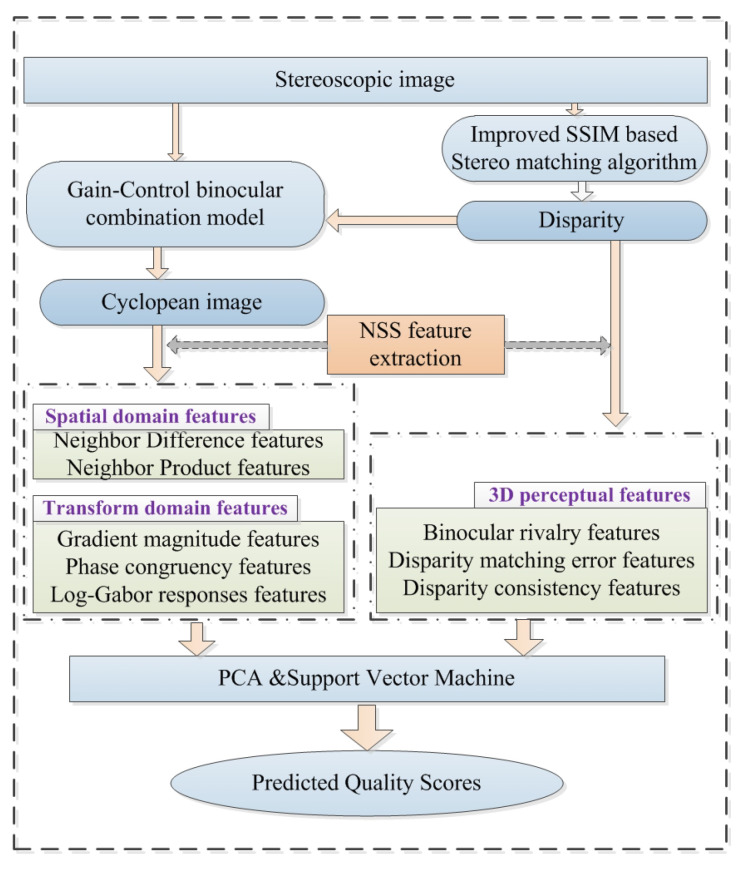
Framework of the proposed NR-SIQA algorithm.

**Figure 2 sensors-23-06230-f002:**
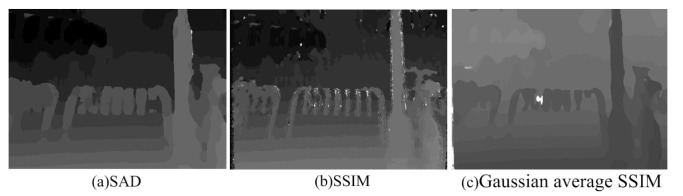
Disparities estimated by different disparity estimation algorithms: (**a**) SAD-based algorithm, (**b**) SSIM-based algorithm, (**c**) Gaussian average SSIM-based algorithm.

**Figure 3 sensors-23-06230-f003:**
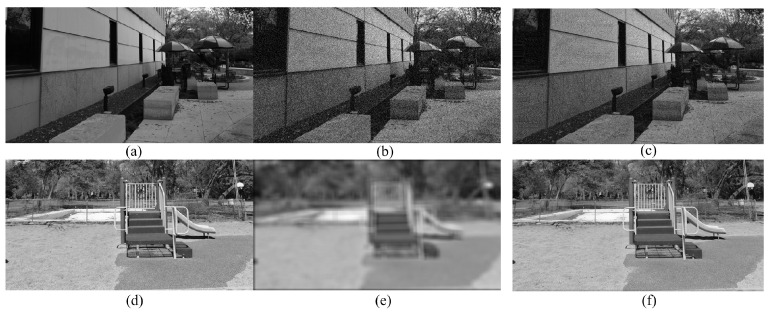
Cyclopean images synthesized by distorted stereo-pairs. (**a**,**b**) and (**d**,**e**) are left and right views of stereo-pairs, respectively. (**c**,**f**) are their respective cyclopean images.

**Figure 4 sensors-23-06230-f004:**
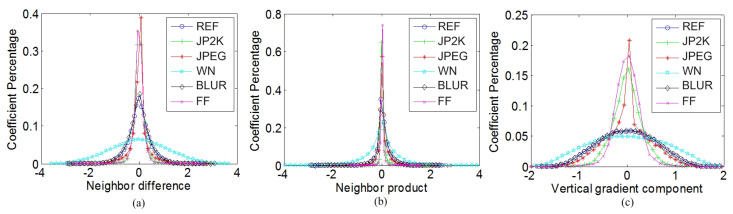
Probability density distributions of (**a**) neighbor difference, (**b**) neighbor product and (**c**) vertical gradient magnitude for six natural cyclopean images synthesized by a reference stereo-pair and its five distorted versions.

**Figure 5 sensors-23-06230-f005:**
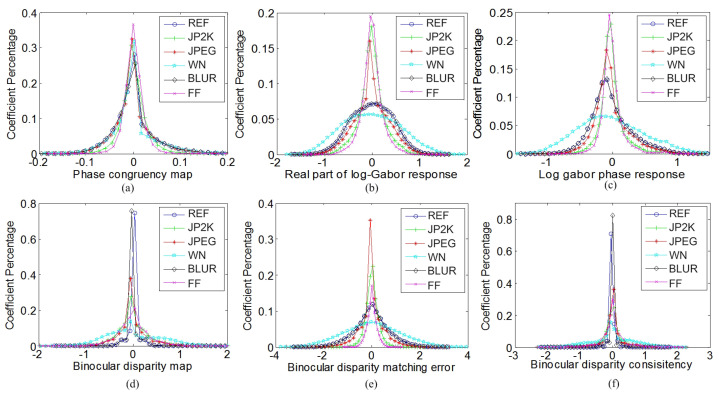
Probability density distributions of (**a**) phase congruency, (**b**) real part of log-Gabor response, (**c**) log-Gabor phase response, (**d**) binocular disparity, (**e**) binocular disparity matching error and (**f**) binocular disparity consistency computed from a reference stereo-pair and its five distorted versions.

**Figure 6 sensors-23-06230-f006:**
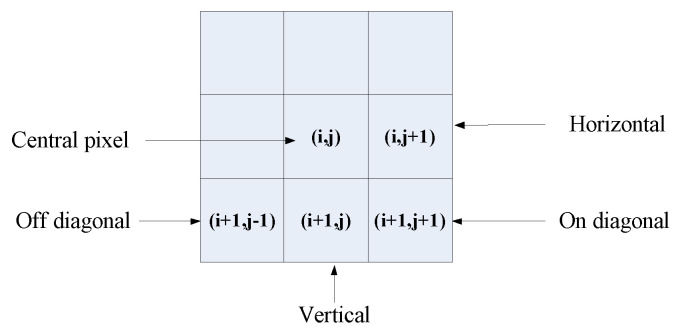
Various pairwise differences computed to quantify neighboring pixel statistical correlation. Neighboring pixel differences are computed along four orientations—horizontal (H), vertical (V), main(on)-diagonal (D1) and secondary(off)-diagonal (D2).

**Figure 7 sensors-23-06230-f007:**
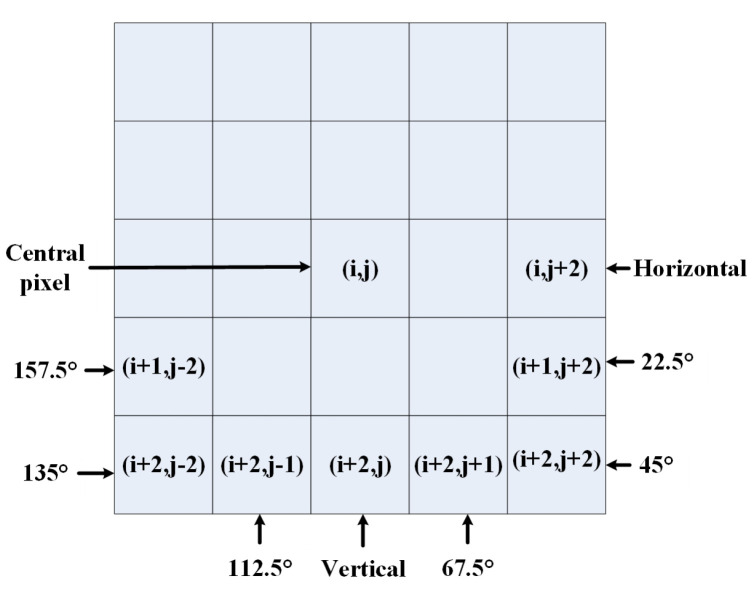
Various pairwise products computed to quantify neighboring pixels’ statistical correlations. Neighboring pixel products are computed along eight orientations—$0^{\circ }$, $22.5^{\circ }$, $45^{\circ }$, $67.5^{\circ }$, $90^{\circ }$, $112.5^{\circ }$, $135^{\circ }$ and $157.5^{\circ }$—at a distance of 2 pixels from central pixel.

**Figure 8 sensors-23-06230-f008:**
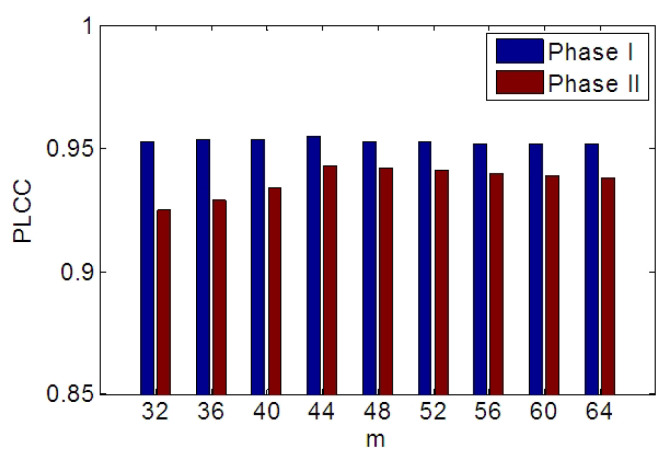
PLCC of the proposed metric with various feature dimensions from LIVE Phase I and Phase II.

**Figure 9 sensors-23-06230-f009:**
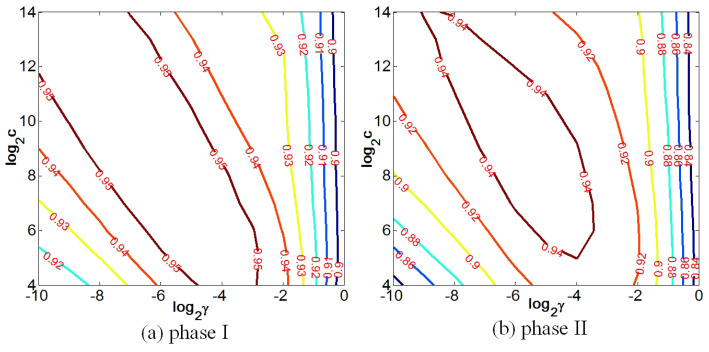
The SVR parameters’ (*C*, γ) selection process on (**a**) LIVE 3D Database Phase I and (**b**) Phase II. The number on the level contour denotes the SROCC value of the cross-validation.

**Table 1 sensors-23-06230-t001:** The categorization of SIQA studies according to the considered visual characteristics and some implementation details.

Algorithm	CYC	NSS	BD	BF	BR	G/P	RM	SD	TD
Gorley [22]	X	X	X	X	X	G	-	O	X
You [35]	X	X	O	X	X	G	-	O	X
Benoit [36]	X	X	O	X	X	G	-	O	X
Chen [38]	O	X	O	O	O	G	-	O	X
Chen [14]	O	O	O	O	O	G	SVR	O	X
Akhter [41]	X	X	O	X	X	P	-	O	X
Ryu [42]	X	X	X	O	O	G	-	O	O
Wang [45]	X	X	O	X	X	P	SVR	X	O
Shao [27]	X	X	O	O	O	P	-	X	O
Shao [43]	X	X	O	O	O	P	-	X	O
Shao [44]	X	X	X	X	X	P	SVR	X	O
Fang [46]	O	O	O	X	O	G	NN	O	X
Karimi [47]	O	O	O	O	X	G	NN	O	X
Proposed	O	O	O	O	O	G	SVR	O	O

**Table 2 sensors-23-06230-t002:** Performance on the training set and testing set as the regression model learned from the **TRAINING** set and **TESTING** set, respectively.

Database	LIVE Phase I			LIVE Phase II		
**Training/Testing**	**SROCC**	**PLCC**	**RMSE**	**SROCC**	**PLCC**	**RMSE**
MTrain→Training set	0.978	0.979	3.265	0.971	0.976	2.452
MTrain→Testing set	0.948	0.958	4.807	0.933	0.939	3.929
MTest→Training set	0.948	0.956	4.826	0.934	0.940	3.881
MTest→Testing set	0.977	0.980	3.282	0.971	0.976	2.482

**Table 3 sensors-23-06230-t003:** Performance on training set and testing set using a regression model learned from the features of a training set with Gaussian white noise added or with no noise.

Database	LIVE Phase I			LIVE Phase II		
**Training/Testing**	**SROCC**	**PLCC**	**RMSE**	**SROCC**	**PLCC**	**RMSE**
WN→Training set	0.962	0.967	4.234	0.950	0.954	3.386
WN→Testing set	0.944	0.954	4.956	0.924	0.936	4.000
WN/F→Training set	0.976	0.978	3.401	0.969	0.973	2.259
WN/F→Testing set	0.952	0.962	4.493	0.940	0.950	3.546

**Table 4 sensors-23-06230-t004:** Performance on LIVE 3D database Phase I and II compared with the existing NR SIQA methods.

Database	LIVE Phase I			LIVE Phase II		
**Algorithm**	**SROCC**	**PLCC**	**RMSE**	**SROCC**	**PLCC**	**RMSE**
Akhter [41]	0.383	0.626	14.827	0.543	0.568	9.249
Chen [14]	0.891	0.895	7.247	0.880	0.895	5.102
Wang [45]	0.828	0.885	7.238	0.794	0.784	7.326
Shao [27]	0.894	0.899	-	-	-	-
Shao [44]	0.950	0.957	-	-	-	-
Fang [46]	0.932	0.936	-	0.931	0.936	-
**Proposed **	**0.952**	**0.962**	**4.493**	**0.940**	**0.950**	**3.546**

**Table 5 sensors-23-06230-t005:** Performance on Waterloo IVC 3D database Phase I and II compared with the existing FR and NR SIQA methods.

Database	UW/IVC Phase I		UW/IVC Phase II	
**Algorithm**	**SROCC**	**PLCC**	**SROCC**	**PLCC**
You [35]	0.597	0.713	0.587	0.682
Chen [38]	0.682	0.734	0.578	0.613
Mittal [58]	0.845	0.869	0.794	0.849
Chen [14]	0.708	0.715	0.547	0.551
**Proposed**	**0.906**	**0.919**	**0.852**	**0.863**

**Table 6 sensors-23-06230-t006:** Performance on each type of distortion on LIVE 3D database Phase I and Phase II.

	Database	LIVE Phase I						LIVE Phase II					
	**Algorithm**	**JP2K**	**JPEG**	**WN**	**BLUR**	**FF**	**ALL**	**JP2K**	**JPEG**	**WN**	**BLUR**	**FF**	**ALL**
	Benoit [36]	0.910	0.603	0.930	**0.931**	0.699	0.899	0.751	0.867	0.923	0.455	0.773	0.728
	You [35]	0.860	0.439	0.940	0.882	0.588	0.878	0.894	0.795	0.909	0.813	0.891	0.786
	Gorley [22]	0.015	0.569	0.741	0.750	0.366	0.142	0.110	0.027	0.875	0.770	0.601	0.146
	Chen [38]	0.888	0.530	0.948	0.925	0.707	0.916	0.814	0.843	**0.940**	0.908	0.884	0.889
SROCC	Shao [27]	0.900	0.607	0.926	0.924	-	0.894	-	-	-	-	-	-
	Shao [44]	0.936	0.818	0.935	0.927	0.814	0.950	-	-	-	-	-	-
	Akhter [41]	0.914	**0.866**	0.675	0.555	0.640	0.383	0.714	0.724	0.649	0.682	0.559	0.543
	Chen [14]	0.919	0.863	0.617	0.878	0.652	0.891	**0.950**	0.867	0.867	0.900	0.933	0.880
	**Proposed**	**0.937**	0.779	**0.959**	0.921	**0.851**	**0.952**	0.946	**0.903**	0.831	**0.912**	**0.939**	**0.940**
	Benoit [36]	0.939	0.640	0.925	0.948	0.747	0.902	0.784	0.853	0.926	0.535	0.807	0.748
	You [35]	0.877	0.487	0.941	0.919	0.730	0.881	0.905	0.830	0.912	0.784	0.915	0.800
	Gorley [22]	0.485	0.312	0.796	0.852	0.364	0.451	0.372	0.322	0.874	0.934	0.706	0.515
	Chen [38]	0.912	0.603	0.942	0.942	0.776	0.917	0.834	0.862	**0.957**	0.963	0.901	0.900
PLCC	Shao [27]	0.872	0.597	0.916	0.923	-	0.899	-	-	-	-	-	-
	Shao [44]	0.949	0.796	0.938	**0.986**	0.837	0.957	-	-	-	-	-	-
	Akhter [41]	0.904	0.905	0.729	0.617	0.503	0.626	0.722	0.776	0.786	0.795	0.674	0.568
	Chen [14]	0.917	**0.907**	0.695	0.917	0.735	0.895	0.947	0.899	0.901	0.941	0.932	0.895
	**Proposed**	**0.958**	0.801	**0.971**	0.965	**0.883**	**0.962**	**0.974**	**0.922**	0.858	**0.977**	**0.949**	**0.950**
	Benoit [36]	4.426	5.022	6.307	4.571	8.257	7.061	6.096	3.787	4.028	11.763	6.894	7.490
	You [35]	6.206	5.709	5.621	5.679	8.492	7.746	4.186	4.086	4.396	8.649	4.649	6.772
	Gorley [22]	11.323	6.211	10.197	7.562	11.569	14.635	9.113	6.940	5.202	4.988	8.155	9.675
	Chen [38]	5.320	5.216	5.581	4.822	7.837	6.533	5.562	3.865	3.368	3.747	4.966	4.987
RMSE	Shao [27]	-	-	-	-	-	-	-	-	-	-	-	-
	Shao [44]	-	-	-	-	-	-	-	-	-	-	-	-
	Akhter [41]	7.092	5.483	4.273	11.387	9.332	14.827	7.416	6.189	4.535	8.450	8.505	9.249
	Chen [14]	6.433	5.402	4.523	5.898	8.322	7.247	3.513	4.298	3.342	4.725	4.180	5.102
	**Proposed**	**3.938**	**3.908**	**3.912**	**4.055**	**5.812**	**4.493**	**2.575**	**3.773**	**2.843**	**3.207**	**3.606**	**3.546**

**Table 7 sensors-23-06230-t007:** Performance with feature type increasing on LIVE 3D database Phase I and Phase II.

Database	LIVE Phase I			LIVE Phase II		
**Feature Set**	**SROCC**	**PLCC**	**RMSE**	**SROCC**	**PLCC**	**RMSE**
PC	0.868	0.879	7.315	0.736	0.775	7.057
PC+GM	0.893	0.912	6.812	0.793	0.857	5.852
PC+GM+GR	0.903	0.922	6.201	0.832	0.869	5.031
PC+GM+GR+ND	0.931	0.943	5.815	0.874	0.891	4.764
PC+GM+GR+ND+NP	0.938	0.948	5.128	0.907	0.918	4.197
PC+GM+GR+ND+NP+3DI	0.948	0.957	4.914	0.932	0.939	3.867

**Table 8 sensors-23-06230-t008:** Performance on LIVE 3D database Phase II training on Phase I.

Criteria	JP2K	JPEG	WN	BLUR	FF	ALL
SROCC	0.864	0.573	0.888	0.883	0.872	0.802
PLCC	0.859	0.581	0.908	0.959	0.889	0.822
RMSE	5.029	5.967	4.625	3.948	5.261	6.427

**Table 9 sensors-23-06230-t009:** Performance on LIVE 3D database Phase I training on Phase II.

Criteria	JP2K	JPEG	WN	BLUR	FF	ALL
SROCC	0.886	0.540	0.879	0.915	0.787	0.867
PLCC	0.928	0.573	0.889	0.944	0.835	0.872
RMSE	4.843	5.359	7.618	4.779	6.845	8.014

**Table 10 sensors-23-06230-t010:** SROCC on LIVE 3D database Phase I for cross-distortion performance evaluation.

Test	JP2K	JPEG	WN	BLUR	FF	ALL
Train						
JP2K	-	0.758	0.911	0.936	0.664	0.898
JPEG	0.586	-	0.926	0.764	0.498	0.746
WN	0.805	0.552	-	0.888	0.724	0.848
BLUR	0.878	0.553	0.921	-	0.737	0.889
FF	0.838	0.470	0.933	0.884	-	0.874

**Table 11 sensors-23-06230-t011:** SROCC on LIVE 3D database Phase II for cross-distortion performance evaluation.

Test	JP2K	JPEG	WN	BLUR	FF	ALL
Train						
JP2K	-	0.757	0.945	0.754	0.906	0.793
JPEG	0.943	-	0.812	0.693	0.700	0.758
WN	0.863	0.588	-	0.621	0.833	0.778
BLUR	0.945	0.386	0.922	-	0.883	0.818
FF	0.930	0.782	0.922	0.841	-	0.887

**Table 12 sensors-23-06230-t012:** Performance evaluation of random database partitioning and image content-based partitioning.

Database	LIVE Phase I			LIVE Phase II		
**Partition Method**	**SROCC**	**PLCC**	**RMSE**	**SROCC**	**PLCC**	**RMSE**
Random-based partition	0.952	0.962	4.493	0.940	0.950	3.546
Content-based partition	0.908	0.928	6.752	0.867	0.866	6.685

**Table 13 sensors-23-06230-t013:** Performance evaluation of different disparity estimation algorithms.

Database	LIVE Phase I			LIVE Phase II		
**Algorithm**	**SROCC**	**PLCC**	**RMSE**	**SROCC**	**PLCC**	**RMSE**
SAD-based	0.952	0.960	4.581	0.927	0.941	3.846
SSIM-based	0.946	0.956	4.841	0.937	0.945	3.727
**Improved SSIM-based**	**0.952**	**0.962**	**4.493**	**0.940**	**0.950**	**3.546**

**Table 14 sensors-23-06230-t014:** Complexity analysis for the proposed method.

Steps	Time Ratios
SSIM-Based Matching and Disparity Generation	36.37%
Cyclopean Generation	2.48%
Spatial Domain Features	1.95%
Transform Domain Features	7.42%
3D Perceptual Features	0.66%
PCA and SVM	51.12%

## Data Availability

Not applicable.

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
