# Peer review of "A Novel No-Reference Quality Assessment Metric for Stereoscopic Images with Consideration of Comprehensive 3D Quality Information"

_sensors, 2023, doi:10.3390/s23136230_

Round 1
Reviewer 1 Report
This paper presents a proposal for a perceptual quality assessment metric for stereoscopic images. I have a number of review remarks related to the manuscript submitted as follows:
* The work presented here is elaborate, which is good. It has a review style presentation which step-by-step attempts to explain the work presented, albeit at too academic detail at times. However, what is not so standing out is that what the unique research novelties made over and above the state-of-the-art methods are. I think a sub-section to be added to the paper on this point, possibly within the Introduction section – there are hints at those but they need to be clearly brought out.
* Following from the remark above, my understanding is that the relevance of already existing methods and techniques has been first proven and then they are used in variety of combinations, and presented as proposed work – this is of course fine, but again novelty is limited – what is the single most important thing that you have devised to achieve your target here?
* Please add the meanings of the abbreviations used in Table 1 (page 3) into the actual table - not easy to find in the text.
* Page 13; Section 3.4: "SVR parameters C and gamma are set to 512 and 0.015625, respectively". Explain why and how? How are they determined? In fact, throughout the paper, please add justifications to all such selected parameters – why and how!?
* Page 14: Not "video quality exports group" but "video quality experts group".
* Table 5 (page 17): RMSE values??
* Table 6 (page 18): How do you explain inferior performance results in the case of BLUR (PLCC), JPEG (PLCC), JP2K (SROCC), WN (both SROCC and PLCC), etc?
* Table 6 (page 18): Do you have the scatter plots of the values presented in this table? At least your own results? Please present them.
* Table 7 (page 18): At what cost? You need to include a complexity analysis here - each additional feature set adds value to performance, fine but what is the added cost with them?
* Page 18; Section 4.4.2: How about cross-database tests involving the Waterloo IVC 3D IQA database (including its two phases)? How is the behaviour?
* Pages 21-24; References section and throughout the manuscript: References are not up to date - there are many references in literature on this very topic, which has been sufficiently mature by now. Please check the recent past! Also, the references cited in the paper do not match the references list - there are citations made which are not included here and vice versa! Not tidy at all.
* A general query: What is the added computational complexity of your presented method with respect to those of taken as benchmarks in literature, as per the performance comparison made against them in Table 6 (page 18)? Please add a section to elaborate on this.
Reviewer 2 Report
There is an important reference paper is missing the references:
https://ieeexplore.ieee.org/abstract/document/8305642
The paper is already from the authors of this paper, but it is not discussed in this recent paper.
The structure of this paper is very similar to the submitted paper,
the comparisons with the previous methods are also in the same order in the above paper as the submitted one.
Please make this issue clear first.
Reviewer 3 Report
In the abstract section, the author should include the quantitative results.
The introduction could be expanded, and more major research sources should be cited.
The relationship between visual information and stereoscopic image perception quality should be covered by the author.
The author should include the methodology framework graphically for this study for a better understanding of the flow of research work.
The author should mention the name of the software/tools used for data analysis.
The author should add more logical justifications and suggestions in the conclusion section.
Round 2
Reviewer 1 Report
Majority of my review comments have been taken into consideration in the revised version. There're still a few outstanding ones, as they do not seem to be addressed and/or at least responded in your response letter.
I believe the paper revision is on track for completion, but please do re-consider my remaining comments from the previous time or if cannot be considered please provide reasons:
1. Table 5: RMSE values??
2. Table 6: Can you explain inferior performance results in the case of BLUR (PLCC), JPEG (PLCC), JP2K (SROCC), WN (both SROCC and PLCC), etc?
3. Table 6: Do you have the scatter plots of the values presented in this table? At least your own results? Please present them.
4. Table 7: At what cost? You need to include a complexity analysis here - each additional feature set adds value to performance, fine but what is the added cost with them?
5. Section 4.4.2: How about cross-database tests involving the Waterloo IVC 3D IQA database (including its two phases)? How is the behaviour?
6. In general: What is the added computational complexity of your presented method with respect to those of taken as benchmarks in literature, as per the performance comparison made against them in Table 6? Please add a section to elaborate on this.
Reviewer 2 Report
The authors listed the differences between two papers, but they still do not cite that paper properly. Since there are large overlaps between two paper (in my opinion), I recommend to give differences and improvements on the manuscript.
